TOPICAL REVIEW

# The glia-immune network: Astrocytes and oligodendrocytes as microglial co-ordinators in health and disease

Verity F. T. Mitchener[1,2] , Millie J. Thackray[1,2] and I. Lorena Arancibia-Cárcamo[1,2]

[1] *UK Dementia Research Institute at UCL, University College London, London, UK*
[2] *The Francis Crick Institute, London, UK*

Handling Editors: Nathan Schoppa & Valentina Mosienko

The peer review history is available in the Supporting Information section of this article (https://doi.org/10.1113/JP287015#support-information-section).

**Abstract figure legend** The glia-immune network involves the communication(s) of glia via immune signalling pathways. In neuro-homeostasis these pathways have been shown to modulate key processes such as synaptic pruning and myelination. The dynamics of these processes is critical to CNS function. In neuropathology, for instance, with amyloid plaque deposition as seen in Alzheimer's disease, there are prominent alterations in the glia-immune network. These alterations induce key pathological changes, including aberrant synaptic pruning and demyelination. These likely have deleterious effects on brain connectivity and function. This review examines the current evidence for the glia-immune network and its relevancy in health and disease.

V. F. T. Mitchener and M. J. Thackray have contributed equally to this work.

**Abstract**  It has long been established that microglia are integral to the CNS immune system. Their surveying and adaptive nature is key in brain development and maintaining homeostasis as well as in the manifestation and progression of neuropathology. However with advancing technology it is becoming increasingly recognised that they do not serve this role in isolation. Previously most work has focused on microglia-derived signalling, with less attention on the sensing and signalling capacity of macroglia (astrocytes, oligodendrocytes). Recent developments in single-cell transcriptomics have allowed extensive analysis of cell profiles in health and disease; these studies have drawn attention to the capacity of macroglia to also engage in immune signalling pathways. This is particularly relevant in neuropathologies, including in Alzheimer's disease (AD), where specific disease-associated profiles of glia (DAGs) have been established. These changes are predominantly related to immune pathways, which were long considered limited to immune cells, including cytokine and chemokine production, antigen presentation and phagocytosis. There is an increasing body of evidence that glia should be considered as active components of the CNS immune system forming a glia-specific immune-like network, whereby macroglia, acting as sensors of the CNS microenvironment, function within this network to co-ordinate diverse CNS effect(s)/function(s). To gain an in-depth understanding of AD pathology, the intimate molecular dialogue of glia needs to be elucidated. This review aims to examine the evidence for macroglia-derived immune signalling and its relevance in health and disease.

(Received 22 January 2025; accepted after revision 20 May 2025; first published online 9 June 2025)

**Corresponding author** I. Lorena Arancibia-Cárcamo: UK Dementia Research Institute at UCL, University College London, London WC1E 6BT, UK.    Email: lorena.arancibia@crick.ac.uk

## Introduction

Glial cells constitute the non-neuronal compartment of the CNS and can be subdivided into microglia, the resident macrophage-derived immune cells, and macroglia, neurally derived cells such as astrocytes and oligodendrocytes. Historically viewed as the structural 'glue' of the brain, glia are now recognised as dynamic cells that serve critical and diverse roles in CNS formation and function (Allen & Lyons, 2018). Macroglia form specialised interactions with neurones – such as tripartite and axo-myelinic synapses – placing them at the forefront of neuronal surveillance. Importantly these cells do not act in isolation but instead participate in an integrative glial network. Through reciprocal communication macroglia and microglia modulate each other's state and function, co-ordinating responses that are central to CNS homeostasis.

One critical means of communication, and the focus of this review, involves immune-related signalling pathways. Traditionally, immune signalling has been associated with inflammatory-mediated pathology; however these pathways are increasingly recognised for their role in the development and maintenance of the CNS (Morimoto & Nakajima, 2019). Moreover dysregulation of these processes has been implicated in neuropathologies, including Alzheimer's disease (AD).

AD is the most prevalent dementia-causing disease, characterised by amyloid-beta (A$\beta$) plaques and tau pathology. Recent work, utilising genome-wide association studies (GWAS) and single-cell sequencing, has identified risk genes predominantly expressed in

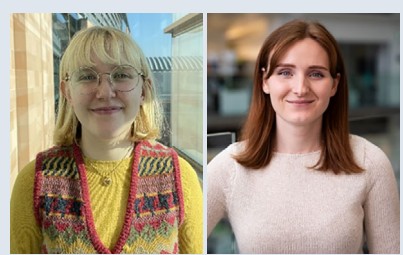

**Verity F. T. Mitchener** holds a PhD from the University of Plymouth where she worked with Robert Fern to model white matter disease and identify myelin-protective therapeutics. She then moved to the UK DRI (UCL) to work with I. Lorena Arancibia-Cárcamo and Bart De Strooper to examine the role of glia in Alzheimer's disease. Verity is interested in illuminating the role of oligodendrocytes and myelination in brain health and disease. **Millie J. Thackray** is a PhD student at UCL investigating glial cross-talk in Alzheimer's disease, with a focus on astrocyte–microglia signalling. She works at the UK Dementia Research Institute at UCL and the Francis Crick Institute under the supervision of I. Lorena Arancibia-Cárcamo and Bart De Strooper. With a BSc in biological sciences and an MRes in translational neuroscience, her research interests lie in neuroimmunology and the role of glia in brain health and neurodegeneration.

microglia and has defined disease-associated cellular substrates across different glial types (Hansen et al., 2018). Such work has led to an increased appreciation for the 'cellular phase' of AD, which occurs prior to the onset of neurodegeneration (De Strooper & Karran, 2016). Spatial transcriptomics approaches have revealed that interglial communication is perturbed around plaques, in particular, their inflammatory signalling pathways (Chen et al., 2020; Mallach et al., 2024). It is therefore clear that understanding how these cellular networks are altered is key in delineating the neuroinflammatory cascade preceding AD symptoms.

Although much of the research on immune signalling in the brain has traditionally centred on how microglia influence neuronal and macroglial functions, emerging evidence highlights the bidirectional and integrative nature of these interactions. Astrocytes and oligodendrocytes play pivotal roles in modulating microglial activity and functions, forming a co-ordinated glial network essential for maintaining CNS homeostasis. This review aims to highlight these underexplored macroglia-microglia pathways and their functional outputs using synapse and myelin dynamics as critical examples, as well as emphasise their disruption in the context of amyloid pathology. Understanding these interactions is critical to uncovering the mechanisms of neuroinflammation and identifying potential therapeutic targets in neurodegenerative diseases.

## Astrocyte-mediated co-ordination of interglial immune signalling pathways

Over the past two decades there has been an increased recognition for the role of astrocytes as active participants in immune signalling pathways (Sofroniew, 2020). Astrocytes have been shown to express an arsenal of pattern-recognition receptors (PRRs), including toll-like receptors (TLRs, e.g. TLR4), C-type lectin receptors (e.g. CLEC9A) and nucleotide-binding oligomerisation domain-like receptors (NLRs, e.g. NLRP3), highlighting their ability to sense their environment (Sofroniew, 2020). Moreover they initiate and engage in immune signalling, releasing alarmins and pro-inflammatory cytokines that alter the microglial immunogenic profile in both physiological and pathological contexts (Table 1). While we focus on AD in this review, many of the molecules presented in Table 1 are implicated more broadly in neuroinflammation, a hallmark pathology of various neurodegenerative diseases. It is therefore likely that the pathways discussed here are relevant across a range of neuropathologies beyond AD.

Early work showed that astrocyte-conditioned media (ACM) is critical for microglial survival and phagocytosis (Bohlen et al., 2017; Min et al., 2006). Since then, astrocytes have been shown to secrete various molecules that regulate this glial network to maintain homeostasis (see Table 1). These include cytokines such as interleukin (IL)-15, IL-3 and IL-$1\beta$, which co-ordinate inflammatory responses (McAlpine et al., 2021; Shi et al., 2020; Wang et al., 2021), and alarmins such as ATP, which shapes microglial injury response(s) (Chen et al., 2024). Depending on their cellular phenotype astrocytes can propagate inflammation by engaging in complement signalling, or they can maintain microglia in a homeostatic state by secreting small molecules, such as the acute-phase reactant protein ORM2 (orosomucoid-2), to interfere with chemokine receptor–ligand interactions (Jo et al., 2017). The dialogue between astrocytes and microglia inevitably has knock-on effects for neuronal networks; we therefore discuss the functional implications of these signalling pathways in physiology and how they are perturbed in amyloid pathology, using the regulation of microglia-mediated synapse pruning as a critical example (Fig. 1).

## Astrocytic co-ordination of microglia-dependent synapse pruning in physiology

Elimination of excess synapses is essential for network development, and our understanding of astrocytes in this process has progressed in recent decades. Although synapse pruning is generally understood to be a microglia-driven process, dependent on a balance of 'eat-me' and 'don't eat me' signals, the appearance of astrocytes in neurodevelopment coincides with periods of synapse pruning, and astrocyte-derived molecules have been shown to modulate synapse elimination by microglia.

An example of this modulation is the alarmin IL-33. Vainchtein and colleagues demonstrated that astrocytes drive cytokine signalling to regulate microglial synapse pruning in development (Vainchtein et al., 2017). They identified astrocytic IL-33, one of the key distinguishing markers of astrocytes from neural progenitors in the developing brain, as essential for recruiting microglia to eliminate excess synapses. The absence of IL-33 leads to dysfunctions in neuronal circuitry, underscoring its role in shaping functional neural networks. Subsequent studies have explored how IL-33 influences downstream microglial responses to explain their increase in synaptic phagocytosis. Astrocyte-derived IL-33 has been shown to promote microglia phagocytosis and bioenergetics through the ST2-AKT pathway during early development (He et al., 2022). Additionally supraphysiological doses of IL-33 induce changes in the microglial expression of MARCO (macrophage receptor with collagenous structure), a PRR typically associated with immune responses to pathogens and debris clearance (Han et al.,

**Table 1. Astrocytic immune molecules and their effect on microglial function in health and disease.**

| Astrocytic immune molecule(s) | Context(s) observed | Effect on microglia | Functional implication | Reference |
|---|---|---|---|---|
| ATP | Injury and disease | Injury and disease response, calcium response | Regulating injury response. | Chen et al. (2024) |
| C3 | Alzheimer's disease | Alters phagocytosis | Acute C3 stimulation promoted and chronic stimulation attenuated microglial phagocytosis *in vitro*. *In vivo* it worsened A$\beta$ pathology. | Lian et al. (2016) |
| PTX3 | Inflammation | Alters phagocytosis | Promoted phagocytosis of zymosan particles and inhibited uptake of apoptotic cells; attenuation of macrophage-mediated phagocytosis of damaged neurones. | Jeon et al. (2010), Ko et al. (2012) |
| TGF-$\beta$ | Inflammation | Anti-inflammatory | IL-10-stimulated astrocytes produce TGF-$\beta$ to attenuate microglial activation (reduced IL-1$\beta$, increased CX3CR1 and IL-4ra). | Norden et al. (2014) |
| IP-10 (CXCL10) | MS | Pro-inflammatory | Migration and activation of microglia in demyelinating lesions. | Tanuma et al. (2006) |
| IL-33 | Development | Alters phagocytosis | Increased microglial synapse engulfment. | Vainchtein et al. (2017) |
| C8 | Alzheimer's disease | Anti-inflammatory | Improves cognitive decline in acute and chronic animal models of AD, ameliorates glial hyperactivation and neuroinflammation. | Kim and Suk (2023) |
| IL-3 | Alzheimer's disease | Alters phagocytosis | Enhances microglial phagocytic capacity of A$\beta$ and motility to prevent cognitive decline and pathology. | McAlpine et al. (2021) |
| CCL2 | Spinal cord injury | Pro-inflammatory | Increases microglial activation and subsequent IL-1$\beta$ release to promote neuronal apoptosis. | Rong et al. (2021) |
| IL-1$\beta$, TNF-$\alpha$ + Nitric oxide | 1,2 DCE induced neuroinflammation | Pro-inflammatory | Increased microglial polarisation. | Wang et al. (2021) |
| IL-15 | Intracerebral haemorrhage | Pro-inflammatory | Worsens neurological deficits and oedema after ICH; increases microglial accumulation close to astrocytes in perhaematomal tissues. Microglial CD86, IL-1$\beta$ and TNF-$\alpha$ increased. | Shi et al. (2020) |

*(Continued)*

**Table 1. (Continued)**

| Astrocytic immune molecule(s) | Context(s) observed | Effect on microglia | Functional implication | Reference |
|---|---|---|---|---|
| CRAMP | EAE and MS | Pro-inflammatory | Increases glial activation and demyelination in EAE. | Bhusal et al. (2022) |
| CXCL12 | *In vitro* | Increased microglial activation | Enhances microglial IL-6 production. | Lu et al. (2009), Rostasy et al. (2003) |
| MFG-E8 | Alzheimer's disease | Increased microglial synapse engulfment | Worsens microglial-dependent synapse loss. | Sokolova et al. (2024) |
| ORM2 | Inflammation | Anti-inflammatory | Modulates microglial inflammation and migration. | Jo et al. (2017) |

$A\beta$, amyloid-beta; AD, Alzheimer's disease; CRAMP, Cathelicidin-related antimicrobial peptide; EAE, Experimental autoimmune encephalomyelitis; IFN, Interferon; IL, interleukin; ICH, Intracerebral haemorrhage; MFG-E8, milk factor globule epidermal growth factor 8; MS, Multiple Sclerosis; ORM2, orosomucoid-2; ST2, Supression of tumorigenicity 2; TGF-$\beta$, transforming growth factor beta; 1,2 DCE, 1,2-dichloroethane.

2023). Both MARCO and ST2 deficiency led to impaired synaptic engulfment and increased seizure susceptibility, a feature also observed in response to knockdown of IL-33 (Han et al., 2023). These findings underscore the importance of astrocyte–microglia communication in the precise regulation of synapse elimination and the establishment of neural circuits during development but may also provide insight into how dysregulation of this communication may occur in neurodegenerative diseases.

Astrocytes also secrete ATP, a neuromodulator involved in regulating synaptic homeostasis, and an alarmin, which co-ordinates injury response in glial and neuronal networks (Cserep et al., 2020). Yang and colleagues highlighted the role of astrocytic ATP signalling in developmental synapse pruning by demonstrating that disruption of astrocytic calcium signalling using ITPR2 receptor knockout impaired developmental synapse elimination. This phenotype was reversed upon administration of ATP and found to be dependent on P2YR1 receptor signalling (Yang et al., 2016). This highlights the role of astrocytic purinergic signalling in the regulation of developing neuronal networks. Microglia have been shown to respond to astrocytic ATP and express an arsenal of purinergic receptors, for example, P2YR12 (Chen et al., 2024; Walker et al., 2020). It is therefore reasonable to speculate that astrocytic purinergic signalling could act through microglia to co-ordinate synapse pruning; however more research is needed to uncover specific mechanisms for this in development.

The complement pathway is one of the main regulators of phagocytosis in monocytes. Microglia are no exception to this, with complement proteins being utilised in neurodevelopment to tag synapses for microglia-mediated elimination (for a detailed review of the complement system in synapse pruning, see Soteros & Sia, 2022). Complement-mediated synaptic pruning during development is essential for correct circuitry formation. Consequently knockout of C1q, the initiator of the classical complement pathway, has been shown to lead to dysfunctions in excitatory circuitry. Importantly the expression of C1q at synapses has been shown to be regulated by astrocytes by a yet unidentified secreted factor (Stevens et al., 2007). Such work highlights the role of complement in mediating glia–synapse communication in the formation of neuronal networks.

Astrocytes can regulate microglial synapse elimination by engaging in immune signalling pathways during development. This is particularly interesting when it is considered that aberrant synapse loss is a key correlate of AD (Terry et al., 1991), a disease marked by dysregulation of immune signalling pathways (Hansen et al., 2018). It is possible that dysregulation of the pathways which fine-tune neuronal circuitry in development is what drives network dysfunction in disease. Although we

have focused on the role of astrocytic co-ordination of microglial synapse pruning, it should be noted that astrocytes themselves can prune synapses via MEGF10 and MERTK during development (Chung et al., 2013). This is particularly interesting in the context of recent work showing synapses can develop normally in CSFR$^{\Delta FIRE/\Delta FIRE}$ mice that lack microglia entirely (O'Keeffe et al., 2025; Surala et al., 2024). Such observations reinforce the potential compensatory mechanisms of glial cells for one another's functions, highlighting their capacity to act within a network.

### Dysregulation of immune pathways in AD may alter astrocytic regulation of microglia-dependent synapse elimination

Dysregulation of synapse elimination is a classic example of AD pathology, and it is likely that microglial regulation of this process is key to disease progression (Hong et al., 2016). A hallmark of AD is reactive astrocytes, observed early in disease (Fontana et al., 2023); interestingly reactive astrocytes are said to antigenically resemble immature astrocytes observed in development (Wilhelmsson et al., 2006).

Key literature has shown that disease-associated and reactive astrocytes exhibit a significant increase in the complement protein C3 (Liddelow et al., 2017). More recently, transcriptomics studies have shown that C1q RNA is significantly upregulated in astrocytes surrounding plaques in the hippocampus (Mallach et al., 2024), the predominant region affected by synapse loss and neuronal death in AD. This synergises with research showing that A$\beta$ induces astrocytic C3 release (Lian et al., 2015) and that inhibiting this pathway reduced synapse loss (Shi et al., 2017) to implicate astrocytic complement signalling as a suspect involved in aberrant microglia synapse pruning in disease.

Interestingly the immunogenic molecule milk factor globule epidermal growth factor 8 (MFG-E8) has been implicated as an astrocytic modulator of aberrant microglial synapse pruning in App$^{NLGF}$ and App$^{NLF}$

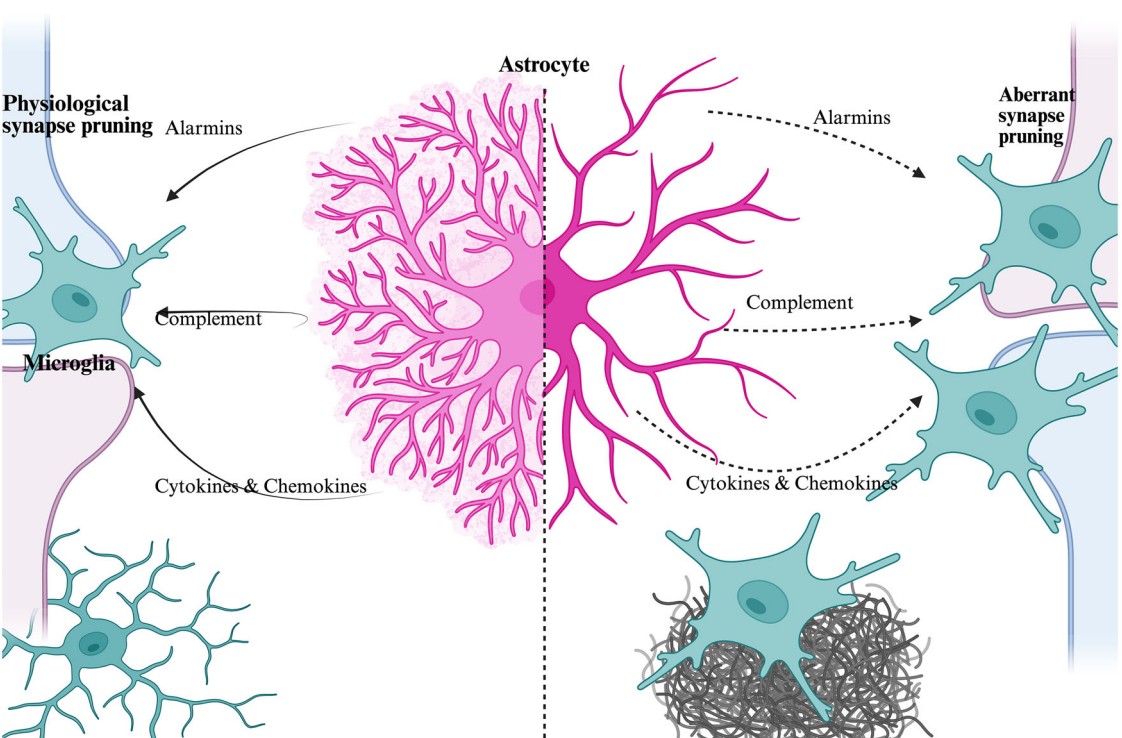

**Figure 1. The role of immune signalling between astrocytes and microglia and their role in synapse pruning**
Astrocyte-derived inflammatory signalling plays a role in regulating microglial synapse pruning in development through the modulation of microglial immunological profiles. In presence of A$\beta$ (amyloid-beta) plaques, astrocyte and microglia phenotypes are perturbed along with such pathways involved in regulating synapse pruning. In AD (Alzheimer's disease) synapse pruning becomes dysregulated, with increased microglial phagocytosis leading to synapse loss. As such, this figure shows the shift of homeostatic to pathological synapse pruning in presence of A$\beta$ plaques.

mouse models (Sokolova et al., 2024). Expression of MFG-E8, a phosphatidyl serine (PtdSer) binding protein predominantly expressed in astrocytes, was shown to be upregulated in a morphologically distinct population of astrocytes present in the hippocampus of App^NLF mice. Furthermore astrocyte-specific knockdown of the receptor prevented microglial synapse engulfment in this disease model, underscoring its role in astrocyte–microglia interactions. Interestingly PtdSer, a canonical 'eat-me' signal that binds to C1q (Paidassi et al., 2008), has been shown to mediate microglial synapse elimination in response to A$\beta$, highlighting a possible convergence of astrocytes and microglia on the PtdSer-C1q pathway during AD pathology (Rueda-Carrasco et al., 2023).

Fractalkine signalling between neuronal CX3CL1 and microglial CX3CR1 during development is important in recruiting microglia to promote excess synapse elimination and circuit maturation (Paolicelli et al., 2011). Whereas in physiological conditions its expression is restricted to neurones, under pathological conditions, in rats, astrocytes have been shown to express high levels of CX3CL1 (Lindia et al., 2005). Dysfunction in this signalling pathway caused by astrocytes may impair synapse maintenance. CX3CL1 has been implicated in AD, but its exact role is unclear. It has been speculated that its secreted form is pro-inflammatory (Bivona et al., 2023; Hulshof et al., 2003), and interestingly it is significantly upregulated in the cerebral spinal fluid (CSF) of AD patients (Bivona et al., 2022). Identifying if this increased secreted CX3CL1 is of astrocytic origin would provide insight into the contribution of astrocytes to dysfunctional fractalkine signalling in disease.

The alarmin pathways astrocytes employ to regulate microglial synapse pruning are also dysregulated in AD; early dysfunction of the ITPR2 receptor, previously mentioned to modulate synapse elimination in a P2YR1-dependent manner, is observed (Kim et al., 2024). Furthermore, microglia exhibit a significant reduction in P2YR12 in AD, possibly hinting at a broader dysregulation of purinergic signalling which may impact synapse elimination (Hansen et al., 2018). Additionally, pathways associated with the alarmin IL-33 are dysregulated in AD. Notably, the downstream signalling molecule of IL-33, AKT, is dysregulated (Soelter et al., 2024), and genetic variants in the IL-33/ST2 pathway are associated with the disease (Lambert et al., 2013).

Regulation of microglial phagocytosis in AD is a double-edged sword. Although we have focused on the potential for astrocyte-derived immune signalling to influence synapse pruning, it is likely the same pathways which impact microglial phagocytosis of A$\beta$ plaques. For example, IL-33 administration in an APP/PS1 mouse model of AD reduced A$\beta$ levels by enhancing microglial phagocytosis, with no effect on synapse elimination (Fu et al., 2016). This may be attributed to the increased expression of the PRR MARCO, which is known to detect A$\beta$ (Brandenburg et al., 2010). Furthermore NFK$\beta$-mediated astrogliosis promoted microglial phagocytosis of A$\beta$ plaque in an APP23 mouse model and was associated with an increase in C3 in both astrocytes and microglia; however the authors did not check if this compromised synapse density (Yang et al., 2021). These examples remind us of the delicate balance between neurodegeneration and neuroprotection which is regulated by this glial network.

## Oligodendrocyte-mediated co-ordination of interglial immune signalling

Literature exploring the capacity of oligodendrocytes to exhibit an immune cell-like signature as well as regulate neuroimmunology has significantly expanded over the past two decades. In fact, it is now becoming increasingly accepted that oligodendrocytes can generally perform immunomodulatory actions and regulate microglial state. Table 2 presents a breakdown of key known molecules involved in oligodendrocyte-mediated regulation of microglia. Key examples are discussed.

Cytokines and chemokines are critical players in immune signalling, and an array of these, including IL-1$\beta$, IL-6, IL-33, TNF, CXCL1, CCL5, CXCL10 and CCL2, are reportedly expressed by oligodendrocytes (Madsen et al., 2020; Pandey et al., 2022; Vela et al., 2002; Watson et al., 2021; Ximerakis et al., 2019). Notably, the secretion of the cytokine IL-33 and the chemokine CCL2 from immunologically active oligodendrocytes has been recently shown to exert direct immunomodulatory function(s), including in the survival, recruitment and responses of microglia (Boccazzi et al., 2021; Norris et al., 2023).

Oligodendrocytes have also been shown to express several neuroprotective, immune-modifying membrane-bound components, including the membrane-bound form of the TNF family (TNFR2). Oligodendrocyte-derived TNFR2 has been shown to hold microglia in a homeostatic state. In fact, the ablation of oligodendrocytic TNFR2 has been shown to accelerate microglial reactivity (Madsen et al., 2020). Oligodendrocytes also express the membrane glycoprotein cluster of differentiation (CD)200, which is thought to act as an immunological 'off-switch' and conveys a neuroprotective effect (Koning et al., 2009). Conversely oligodendrocytes also express immune-activating components on their membranes, such as the Cx3C receptor (CX3CR1) and the CX3CL1 ligand (fractalkine) (Watson et al., 2021). These molecules act as a part of an 'eat-me' cascade and are predominately associated with prompting the phagocytic nature of microglia (Nemes-Baran et al., 2020; Olveda et al., 2024).

**Table 2. Oligodendrocytic immune molecules and their effect on microglial function in health and disease.**

| Oligodendrocytic immune molecule(s) | Context(s) observed | Effect on microglia | Functional implication | Reference |
|---|---|---|---|---|
| Fractalkine | Developmental myelination | Pro-phagocytic Pro-migratory | Phagocytosis of OPCs, regulation of OPC number and myelination | Nemes-Baran et al. (2020) |
| Fractalkine and fractalkine receptor (CX3CR1) | Oligodendrocyte genesis | – | Regulation of oligodendrocyte genesis | Watson et al. (2021) |
| Fractalkine | Neurodegeneration | Pro-phagocytic | Phagocytosis of dying oligodendrocytes | Olveda et al. (2024) |
| IL-33 | Infection | Pro-survival Activation | Neuroprotective | Norris et al. (2023) |
| IL-33 | Brain ageing | – | – | Ximerakis et al. (2019) |
| IL-1β | Development | – | Oligodendrocyte genesis | Vela et al. (2002) |
| Connexin-47 | Experimental autoimmune encephalomyelitis | Immune modulation Anti-inflammatory | Myelin- and neuroprotective | Zhao et al. (2020) |
| CCL2, IFN-β, IL-1β and CXCL10 | Neuroinflammation | Pro-inflammatory | Altered microglial phenotype and function and altered oligodendrocyte genesis | Boccazzi et al.(2021) |
| CXCL10, CCL2, CCL3 and CCL5 | *In vitro* | – | – | Balabanov et al. (2007) |
| TNFR2 | Experimental autoimmune encephalomyelitis | Immune modulation | Neuro- and myelin protective | Madsen et al. (2020) |
| C1q, C1s, C2, C3, C4, C5, C6, C7, C8 | *In vitro* | – | – | Hosokawa et al. (2003) |
| CD59 DAF MCP | *In vitro* | – | – | Gasque and Morgan (1996) |
| CD59 DAF | *In vitro* | – | – | Scolding et al. (1998) |
| CD200 | Multiple sclerosis | Tolerance | Neuroprotective | Koning et al. (2009) |
| SERPINA3, C4b, MHC-1, IL-33 | Alzheimer's disease, 5x FAD mice, experimental autoimmune encephalomyelitis and ageing | – | – | Kenigsbuch et al. (2022) |
| SERPINA, C4b, IL-1β, Tnfrs1a, Hmox1, Bace2, B2m | Mouse models of AD and MS, including PS2APP, TauP301S, TripleTg, cuprizone, and lysolecthin | – | – | Pandey et al. (2022) |

*(Continued)*

**Table 2. (Continued)**

| Oligodendrocytic immune molecule(s) | Context(s) observed | Effect on microglia | Functional implication | Reference |
|---|---|---|---|---|
| C4b, SERPINA | 5xFAD mice | – | – | Zhou et al. (2020) |
| C4b, SERPINA | Mouse models of Alzheimer's disease, including PS2APP, TauP301L, and Tau/APP and Trem2–/– mice | – | – | Lee et al. (2021) |

AD, Alzheimer's disease; CD, cluster of differentiation; DAF, decay-accelerating factor; IL, interleukin; MCP, membrane cofactor protein; OPC, oligodendrocyte precusor cell.

Oligodendrocytes have also been shown to utilise gap junctions such as connexin-47 (Cx47) to facilitate inter-cellular communication to other glia, that is, astrocytes (Wasseff & Scherer, 2011). Although the capacity of glia to communicate via these junctions is well characterised (Moore & O'Brien, 2015), recent work has revealed that this communication can be immune modifying. Targeted Cx47 ablation in oligodendrocytes has been shown to enhance pro-inflammatory gene expression in microglia (Zhao et al., 2020). However, whether this effect is direct or acts as a part of the larger glial network requires further clarification.

Collectively, these findings highlight the multifaceted role of oligodendrocytes in modulating microglial behaviour and maintaining immune balance in the CNS. Additionally it should be noted that most of these immune pathways can actually feedback onto oligodendrocytes to dictate the formation and maintenance of the cell and their hallmark myelin sheaths.

## Oligodendrocytic co-ordination of microglial-dependent myelin dynamics

Our understanding of the processes that regulate and maintain myelination has increased over the past few decades. While the focus has largely been on activity-dependent neuro-signalling, recent work has increasingly highlighted an immune signalling involvement in oligodendrocyte cell cycle and myelination. Key examples are provided in Table 2. This section highlights the key oligodendrocyte-derived immune signalling molecules that act to regulate myelin dynamics via their regulation of key microglial functions (Fig. 2).

Recent works have demonstrated the critical role of microglia in regulating and maintaining myelin. Here depletion of microglia in CSF1R-deficient mice showed myelin deficits (McNamara et al., 2023; Nemes-Baran et al., 2020). Oligodendrocytes in these mice exhibited alterations in key immune signals, including the cytokine transforming growth factor beta (TGF-$\beta$) and the complement component C4b, implying the importance of immune-mediated oligodendrocyte–microglia cross-talk in myelination (McNamara et al., 2023). TGF-$\beta$ has been shown to regulate the timing(s) of myelination, with TGF-$\beta$ receptor depletion in OPCs preventing their maturation and resulting in hypomyelination. The loss of oligodendrocytic TGF-$\beta$ receptors mimicked observations made in the CSF1R-deficient mice, indicating the importance of the TGF-$\beta$ axis in myelination (McNamara et al., 2023).

In addition to regulating oligodendrocyte development, key immune components can regulate oligodendrocyte number and myelination via critical 'eat-me' signals,

which induce the microglial-mediated phagocytosis of oligodendrocytes and/or myelin. Oligodendrocytes have been shown to express the CX3C receptor and ligand (Watson et al., 2021). CX3C signalling is known to be a main regulator of microglial phagocytosis. This regulates not only oligodendrocyte number but also their maturation and myelination status. For example, fractalkine increases OPC genesis and differentiation, with CX3CR1-deficient mice exhibiting impairment of microglial-mediated OPC clearance. Although this results in increased oligodendrocyte numbers, defective myelination occurs (Nemes-Baran et al., 2020; Olveda et al., 2024). It has also been noted that the phagocytosis of OPCs by microglia is dependent on TLR4. This effect is restricted by the immune-privilege protein CD200 expressed on the OPC surface (Hayakawa et al., 2016). CD47 is another surface integrin expressed on myelin, which binds receptors on microglia to down-regulate the engulfment of myelin (Gitik et al., 2011). The phagocytosis of myelin is also regulated by complement

components such as CR3 (Rotshenker et al., 2008; van der Laan et al., 1996). Oligodendrocytes have been shown to express, engage in and be sensitive to complement (Hosokawa et al., 2003; Pandey et al., 2022). In addition the phagocytosis of myelin by microglia has been shown to alter downstream immune signalling, causing TNF-$\alpha$ and nitric oxide production (van der Laan et al., 1996).

## Oligodendrocyte-derived immune signalling as a contributor to A$\beta$ pathology and AD

Recent significant findings indicate the importance of oligodendrocytes in A$\beta$ pathology in AD. It has been demonstrated not only that oligodendrocytes have the capacity to produce A$\beta$ but also that they act as major contributors to A$\beta$ plaque burden (Sasmita et al., 2024). Furthermore it has been observed that specific suppression of oligodendrocyte-derived A$\beta$ rescues hallmark neuronal hyperactivity in the App[NL-G-F] knock-in mouse model (Rajani et al., 2024).

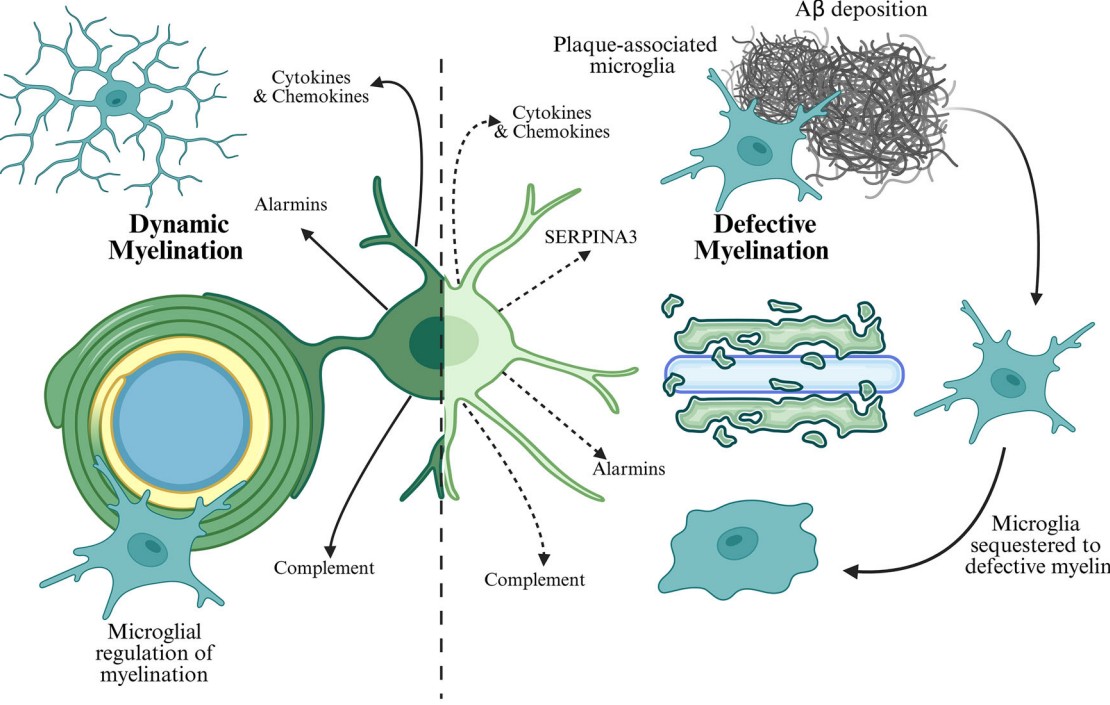

**Figure 2. Immune signalling between oligodendrocytes and microglia and their role in myelin dynamics**
Immune signalling is a key mediator of myelin dynamics, including in the regulation of oligodendrogenesis and myelin formation. Microglia are known to regulate myelin via their phagocytic capacity by clearing myelin and regulating oligodendrocyte cell number and maturation. It is known that oligodendrocytes can engage in immune signalling via the release of factors such as cytokines, chemokines and alarmins as well as complement components. In neuropathology, including in presence of A$\beta$ (amyloid-beta) plaques, oligodendrocytes exhibit a disease-associated profile which converges across several neuropathologies. This involves key alterations in immune signalling, including interferon-stimulated genes, antigen presentation, cytokines and chemokines and complement components. A major factor identified in the response of oligodendrocytes to A$\beta$ is the serine protease inhibitor SERPINA3. In addition, myelin is shown to become defective impairing tissue homeostasis; this can drive plaque deposition and alter microglia profiles and functions. For instance, defective myelin can act as a molecular distraction sequestering microglia away from plaque clearance.

The latest advancements in single-cell profiling have established disease-associated profiles of oligodendrocytes in AD (Lee et al., 2021; Zhou et al., 2020). This profile has also been shown to be consistent across CNS pathologies as well as brain ageing (Kenigsbuch et al., 2022; Pandey et al., 2022). It is defined by changes in key genes relating to neuroinflammation, including interferon-stimulated genes (*Ifit* and *Oas*), antigen presentation genes (*MHC-I* and *MHC-II*), cytokines and chemokines (IL-33, IL-1$\beta$, TNF, TNFRSF12A and CCL4) and complement components (C4b).

A major molecule associated with disease state was deemed to be SERPINA3/SERPINA3N (also known as a1-antichymotrypsin). SERPINA3 is a serine protease inhibitor related to immune proteases and is primarily secreted in response to systemic inflammation. SERPINA3 has demonstrated a mixed profile in CNS pathology exhibiting both neuroinflammatory and neuroprotective effects (Haile et al., 2015; Mucke et al., 2000). A recent and comprehensive review of the complex role of SERPINA3 in neuropathology is available in Zhu et al. (2024). To briefly summarise it is increasingly thought that oligodendrocytes may be the primary source of SERPINA3n in the CNS (Zhu et al., 2024). SERPINA3 has been proposed as a biomarker for CNS pathology, including in AD and multiple sclerosis (Kenigsbuch et al., 2022). Although not exclusive to AD, SERPINA3 has been shown to be significant in AD pathology. For instance it has been shown to accumulate within the amyloid plaques of AD brains and can directly interact with A$\beta$ (Abraham et al., 1988; Zhu et al., 2024). Furthermore SERPINA3 overexpression was sufficient to enhance plaque deposition in humanised APP mice (Mucke et al., 2000).

Interestingly, the intensity of oligodendrocyte response to A$\beta$ pathology is shown to correlate with disease severity and likely with the degree of neuroinflammation (Kenigsbuch et al., 2022). Overall this suggests not only that oligodendrocytes play a significant, and perhaps underestimated, role in A$\beta$ pathology but also that the manner in which oligodendrocytes appear to respond seems to predominately cluster to immune pathways.

## The oligodendrocytic immune role in myelin alterations as a manifestation of AD pathology

Myelin alterations are now considered as primary and primordial in the manifestation of AD pathology (Papuć & Rejdak, 2020; Schoemaker et al., 2022). Recently, it has been shown that myelin dysfunction can drive plaque formation and alter microglial profiles and functions (Depp et al., 2023). Furthermore, the oligodendrocyte–microglia gene networks have recently been demonstrated to be enriched, and even modify

disease risk, in AD (Graham et al., 2025). This identified the preservation of homeostatic microglia as a critical feature of longevity. Interestingly it is known, as discussed earlier, that oligodendrocytes play a critical role in restraining microglia. Whereas heightened disease risk was shown to relate to age-related inflammation, genes enriched in oligodendrocytes in ageing were related to both lipid and cholesterol processing, both of which are critical in myelination. This work also highlighted putative risk genes associated with complement and lysosomes, both of which regulate myelin via pruning and clearance processes. Interestingly, these networks converged between ageing and demyelination, suggesting AD risk is associated with dysregulation of myelin as well as the altered communication between oligodendrocytes and microglia (Fig. 2).

Several cytokines and chemokines have been shown to play a critical role in the regulation of myelination as well as the AD disease state. Key examples include fractalkine, IL-1$\beta$, IL-33 and TGF-$\beta$, all of which have been shown to be released by oligodendrocytes, particularly in disease, to regulate microglial state and function (Boccazzi et al., 2021; Gadani et al., 2015; Madsen et al., 2020; Norris et al., 2023). The role of these factors in myelin injury in AD is complex. Generally these cytokines are considered to have positive effects on myelination, promoting myelin maintenance and repair (Gadani et al., 2015; Zhang et al., 2006). Myelin is thought to become defective in ageing, and the removal and regeneration of myelin is required for dynamic tissue homeostasis (Hill et al., 2018; Lampron et al., 2015; Rawji et al., 2018). These factors have been shown to regulate the proliferation, differentiation and maturation of OPCs, a critical step in the formation of new myelin (McNamara et al., 2023; Vela et al., 2002; Zhang et al., 2006). They also play a critical role in myelin homeostasis; it has been shown that fractalkine is critical in the microglia-mediated clearance of dead oligodendrocytes (Olveda et al., 2024). Fractalkine also regulates the formation of new oligodendrocytes and myelin and promotes regenerative processes (remyelination), with fractalkine receptor-deficient mice exhibiting impaired myelination (de Almeida et al., 2023; Nemes-Baran et al., 2020). It is thought that dysregulation of these processes in ageing impairs tissue homeostasis, and this is a principal means by which myelin dysfunction is thought to drive AD pathology (Fig. 2) (Depp et al., 2023).

However it should be noted that these same factors can also trigger processes which harbour negative effects on myelination, causing oligodendrocyte death and demyelination (Jurewicz et al., 2003; Takahashi et al., 2003). Their role is further complicated in the AD environment, as the phagocytic-inducing capacity of these cytokines is critical in promoting the clearance of A$\beta$ and reducing plaque burden, thus having positive effects for neuropathology (Wyss-Coray et al., 2001). Further work

is required to closely examine how these processes are balanced and the specific mediators for each as well as the specific role of oligodendrocyte immune signalling.

## Conclusion and future perspectives

Here, we proposed a framework by which macroglia, acting as close regulators of neuronal homeostasis, regulate CNS dynamics by engaging in an immune-like network. We provided specific examples of the complex immune signalling pathways used by macroglia to modulate microglial functions such as the phagocytosis of synapses and myelin. Although we have focused on the regulation of these processes, these pathways likely feed into the co-ordination of a larger immune-like network to regulate CNS dynamics in both health and disease. Furthermore, it should be considered that other glia and cell types, not mentioned as a part of this review, likely contribute to these networks and processes. Nonetheless, this review presents increasing evidence for glia and their communications as major contributors to AD pathology and immune pathways as critical to these responses (Mallach et al., 2024). It should be noted that this review also highlights convergent pathways in other neuropathologies. Disease-associated profiles of glia are often shared, and this offers insights into the universality of glia in disease as well as potential underexplored pathways for the AD field. This review has also highlighted key limitations in our existing understanding of macroglia communications. There is currently a lack of understanding of how immune pathways converge and diverge specifically in AD, where processes are confounded by the presence of plaque. We need to delineate the manner in which pathways support $A\beta$ clearance while avoiding aberrant processes, including synapse elimination and myelin loss. Further interrogation of the proteomic alterations in macroglia will elucidate how specific factors shape both homeostatic and neuropathological processes in tandem. Increased mass spectrometry sensitivity and advancements in the proximity labelling toolkit can now be used to interrogate the secretome and interactome of these cells in more detail (Cho et al., 2020). 'Omics' technologies are key in allowing us to dissect complex cellular responses in an AD environment; however, understanding how cellular function is impacted is ultimately key in investigating disease progression, and functional evidence of these processes is limited. Elucidating the functional implications of the immune signalling network between glia in the CNS is critical to better understanding and therefore preventing disease progression in AD. Although we have demonstrated the potential for the significant role of macroglia-derived immune factors in synapse and myelin loss in AD, more direct and comprehensive evidence of the involvement for these signalling pathways is required.

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

## Additional information

### Competing interests

No competing interests declared.

### Author contributions

I.L.A.-C. contributed to the conceptualisation and provided oversight and input throughout all stages of manuscript preparation. Both V.F.T.M. and M.J.T. wrote the abstract, introduction and conclusion and prepared the figures and tables. M.J.T. wrote sections discussing the role of astrocytes and synaptic pruning in health and disease. V.F.T.M. wrote sections discussing the role of oligodendrocytes and myelination in health and disease. All authors have approved the final version of the manuscript and agreed to be accountable for all aspects of the work. All persons designated as authors qualify for authorship, and all those who qualify for authorship are listed.

### Funding

This work was supported by the UK Dementia Research Institute (award number UK DRI-1004), which receives its funding from UK DRI, Ltd, funded by the UK Medical Research Council, Alzheimer's Society and Alzheimer's Research UK. I.L.A.-C. also receives funding as a co-investigator on a Medical Research Council Programme Grant (MR/Y014847/1).

### Acknowledgements

The authors thank James A. Conway for proof-reading and reviewing the manuscript.

### Keywords

Alzheimer's disease, astrocyte, glia, immunology, microglia, myelin, oligodendrocyte, synapse pruning

### Supporting information

Additional supporting information can be found online in the Supporting Information section at the end of the HTML view of the article. Supporting information files available:

**Peer Review History**

