## [Peer Review History · The Journal of Physiology]

The Glia-Immune Network: Astrocytes and Oligodendrocytes as microglial coordinators in health and disease.

Verity Fay Toulouse Mitchener, Millie Jennifer Thackray, and I. Lorena Arancibia-Carcamo
DOI: 10.1113/JP287015

Corresponding author(s): I. Lorena Arancibia-Carcamo (lorena.arancibia@crick.ac.uk)

Review Timeline:

Submission Date:	22-Jan-2025
Editorial Decision:	20-Mar-2025
Revision Received:	06-May-2025
Accepted:	20-May-2025

Senior Editor: Nathan Schoppa

Reviewing Editor: Valentina Mosienko

Transaction Report:

Dear Dr Arancibia-Carcamo,

Re: JP-TR-2025-287015 "**The Glia-Immune Network: Astrocytes and Oligodendrocytes as microglial coordinators in health and disease.**" by Verity Fay Toulouse Mitchener, Millie Jennifer Thackray, and I. Lorena Arancibia-Carcamo

Thank you for submitting your manuscript to The Journal of Physiology. It has been assessed by a Reviewing Editor and by 2 expert referees and we are pleased to tell you that it is acceptable for publication following satisfactory revision.

ABSTRACT FIGURES: Authors may use The Journal's premium BioRender account to create/redraw their Abstract Figures (and any other suitable schematic figure). Information on how to access this account is here: <https://physoc.onlinelibrary.wiley.com/journal/14697793/biorender-access>.

REVISION CHECKLIST: Upload a full Response to Referees file. To create your 'Response to Referees' copy all the reports, including any comments from the Senior and Reviewing Editors, into a Microsoft Word, or similar, file and respond to each point, using font or background colour to distinguish comments and responses and upload as the required file type.

We look forward to receiving your revised submission.

Yours sincerely,

Nathan Schoppa
Senior Editor

REQUIRED ITEMS FOR REVISION

- Your MS must include a complete "Additional information section" with the following 4 headings and content:

Competing Interests: A statement regarding competing interests. If there are no competing interests, a statement to this effect must be included. All authors should disclose any conflict of interest in accordance with journal policy.

Author contributions: Each author should take responsibility for a particular section of the study and have contributed to writing the paper. Acquisition of funding, administrative support or the collection of data alone does not justify authorship; these contributions to the study should be listed in the Acknowledgements. Additional information such as 'X and Y have contributed equally to this work' may be added as a footnote on the title page.

It must be stated that all authors approved the final version of the manuscript and that all persons designated as authors qualify for authorship, and all those who qualify for authorship are listed.

Funding: Authors must indicate all sources of funding, including grant numbers. If authors have not received funding, this must be stated.

It is the responsibility of authors funded by RCUK to adhere to their policy regarding funding sources and underlying research material. The policy requires funding information to be included within the acknowledgement section of a paper. Guidance on how to acknowledge funding information is provided by the Research Information Network. The policy also requires all research papers, if applicable, to include a statement on how any underlying research materials, such as data, samples or models, can be accessed. However, the policy does not require that the data must be made open. If there are considered to be good or compelling reasons to protect access to the data, for example commercial confidentiality or legitimate sensitivities around data derived from potentially identifiable human participants, these should be included in the statement.

Acknowledgements: Acknowledgements should be the minimum consistent with courtesy. The wording of acknowledgements of scientific assistance or advice must have been seen and approved by the persons concerned. This section should not include details of funding.

- Please upload separate high quality figure files via the submission form.

- Author profile(s) must be uploaded via the submission form. Authors should submit a short biography (no more than 100 words for one author or 150 words in total for two authors) and a portrait photograph of the two leading authors on the paper. These should be uploaded and clearly labelled together in a Word document with the revised version of the manuscript. Any standard image format for the photograph is acceptable, but the resolution should be at least 300 DPI and preferably more. A group photograph of all authors is also acceptable, providing the biography for the whole group does not exceed 150 words.

- Please include a full title page as part of your main article (Word) file, which should contain the following: title, authors, affiliations, corresponding author name and contact details, keywords, and running title.

- Please ensure that the Article File you upload is a Word file.

EDITOR COMMENTS

Reviewing Editor:

Thank you for your excellent and timely review, which provides a clear and comprehensive summary of recently published studies on the interaction between glial cells in health and neurodegenerative disease. The reviewers highlighted a few areas of improvement - please incorporate their suggestions into the revised version of the manuscript.

Senior Editor:

Thank you for submitting your review manuscript to Journal of Physiology. It has been examined by two expert referees and reviewing editor, who felt that the review is timely, comprehensive, and generally well-done. The referees each have a number of minor concerns, all of which should be addressed in a revised manuscript. We look forward to seeing your revision.

REFEREE COMMENTS

Referee #1:

The review article by Verity et al. provides a well-structured and insightful summary of recent findings on the network perspective of brain glial cells in both health and disease. The focus on extracellular signals that directly or indirectly modulate microglial function, using Alzheimer's disease (AD) as a pathological model, effectively illustrates how dysregulation of these signals contributes to disease progression. The authors structured the manuscript into two major sections: one addressing astrocyte-microglia interactions and the other focusing on oligodendrocyte-microglia crosstalk. Overall, this review presents a timely and comprehensive perspective, supported by an extensive body of literature on molecular mechanisms and disease relevance. I have only a few minor suggestions that could further enhance the clarity and impact of the manuscript:

1. The central concept of this review—a network-based view of glial cell function—is highly valuable and represents an important step forward from the classical approach that focuses on individual cell types. In light of this, I suggest a slight revision of the title and introduction to better reflect this integrative perspective, as they currently appear somewhat microglia-centric. While the emphasis on signals directed toward microglia is well-justified, a network-based view implies that these inputs dynamically influence microglia, which in turn modulate signals from other glial cells. The authors touch on this idea at several points, but reinforcing it more explicitly would better support the integrative framework they propose. Additionally, the terminology "glia-immune" and "glia-to-microglia" might benefit from reconsideration, as microglia are typically classified as a type of glial cell. Clarifying these terms could help align the language more closely with the overall theme of glial network interactions.
2. On page 8, the discussion of astrocyte-microglia crosstalk in synapse pruning raises a point about the uncertainty in the cell-specific function of P2Y1 in the referenced study. However, this is followed by a statement on P2Y12 expression in microglia as supporting evidence for a potential role of microglia, which may not be entirely logically consistent. While both P2Y1 and P2Y12 are purinergic receptors, their expression patterns and functional roles differ significantly. A clearer distinction between these two receptors, or a slight rewording of this section, would improve the precision of the argument.
3. As the review's central theme is the crosstalk between glial cells, it may enhance clarity to place greater emphasis on pathways that involve intercellular communication, while not mentioning those that only affect individual cell types. For instance, on page 14, between two well-selected examples demonstrating microglial involvement in myelin maintenance and phagocytosis, there is a paragraph discussing IL-1 β secretion from oligodendrocytes that lacks a clear connection to microglial function. Keeping the discussion more tightly focused on cell-cell interactions would strengthen the manuscript's overall coherence and readability.
4. The ordering of physiological and pathological states is somewhat inconsistent in different sections. For example, in the discussion on immune signals regulating microglia-dependent phagocytosis, AD-related changes are described first, followed by physiological mechanisms. However, in other parts of the review, the physiological role is introduced first, followed by disease-related alterations. Maintaining a consistent structure—either "health to disease" or "disease to health"—throughout the manuscript would improve readability and logical flow.
5. The summary tables provide valuable details on secreted factors across multiple disease models, while the main text primarily focuses on AD. It may be helpful to briefly discuss whether these molecules are specific to AD pathology or part of broader neurodegenerative mechanisms. This additional context would allow readers to better understand the generalizability of these findings and their potential relevance to other neurological disorders.

Referee #2:

This review discusses the latest findings on the bidirectional communications between macroglia (astrocytes and oligodendrocytes) and microglia, especially how macroglia regulate and control microglial functions during brain development and in neurological diseases with a focus on AD pathology. This review comprehensively lists and describes recently discovered molecules and pathways that are involved in the interactions between macroglia and microglia, emphasizing the immunomodulatory roles of astrocytes and oligodendrocytes in health and disease, the aspect of which was underappreciated previously in the field. A couple of minor points are as following:

1. Regarding the titles "Astrocytes initiate and engage in immunological signalling pathways" and "Oligodendrocytes initiate and engage in immunological signalling pathways", it might not be very precise to describe astrocytes or oligodendrocytes as the cells that initiate immunological signalling pathway, in a complex multi-cellular system, it's difficult to dissect out which cell types are the initiators, unless in a very specific context.
2. The authors discuss "Astrocytic coordination of microglia-dependent synapse pruning in physiology", some recent studies in the field, e.g. PMID: 39762658, have challenged the previous "established" role of microglia in synapse pruning and circuit formation during brain development, showing normal neuron development without microglia in the brain. These new findings are important, should be discussed or mentioned in this session.

END OF COMMENTS

RE: JP-TR-2025-287015R1

Dear Editor,

Thank you and the reviewers for your positive feedback and helpful suggestions. Below we have summarised point-by-point our response along with a summary of the corresponding changes made to the manuscript.

Referee #1

We thank the reviewer for their positive feedback and acknowledgement that *this review presents a timely and comprehensive perspective.*

1. The central concept of this review—a network-based view of glial cell function—is highly valuable and represents an important step forward from the classical approach that focuses on individual cell types. In light of this, I suggest a slight revision of the title and introduction to better reflect this integrative perspective, as they currently appear somewhat microglia-centric. While the emphasis on signals directed toward microglia is well-justified, a network-based view implies that these inputs dynamically influence microglia, which in turn modulate signals from other glial cells. The authors touch on this idea at several points, but reinforcing it more explicitly would better support the integrative framework they propose. Additionally, the terminology "glia-immune" and "glia-to-microglia" might benefit from reconsideration, as microglia are typically classified as a type of glial cell. Clarifying these terms could help align the language more closely with the overall theme of glial network interactions.

We thank the reviewer for highlighting the conceptual value of this review. In response to their comment, we have reworded the introduction to better emphasise the network-based view of glial function. As the reviewer rightly notes, the primary aim of this review is to specifically highlight the underexplored signalling from macroglia to microglia. We have ensured this is now explicitly stated in the introduction. Additionally, the phrasing “glia to microglia” has been removed and replaced with “inter-glial communication”. However, we have retained the term “glia-immune” in the title as we believe it effectively describes the immunological interactions between all glia, rather than distinguishing between specific cell-types.

2. On page 8, the discussion of astrocyte-microglia crosstalk in synapse pruning raises a point about the uncertainty in the cell-specific function of P2Y1 in the referenced study. However, this is followed by a statement on P2Y12 expression in microglia as supporting evidence for a potential role of microglia, which may not be entirely logically consistent. While both P2Y1 and P2Y12 are purinergic receptors, their expression patterns and functional roles differ significantly. A clearer distinction between these two receptors, or a slight rewording of this section, would improve the precision of the argument.

We apologise for the confusion. This section (page 6/7) has now been amended for clarity to discuss general microglial responses to astrocytic ATP.

3. As the review's central theme is the crosstalk between glial cells, it may enhance clarity to place greater emphasis on pathways that involve intercellular communication, while not mentioning those that only affect individual cell types. For instance, on page 14, between two well-selected examples demonstrating microglial involvement in myelin maintenance and phagocytosis, there is a paragraph discussing IL-1 β secretion from oligodendrocytes that lacks a clear connection to microglial function. Keeping the discussion more tightly focused on cell-cell interactions would strengthen the manuscript's overall coherence and readability.

Thank you to the reviewer for their constructive suggestion. We have revised the relevant section to more tightly emphasise intercellular communication. We have removed the discussion of IL-1 β from the Vela et al., 2002 study. However, we feel it is important to note that oligodendrocytes have the capacity to produce cytokines which are known to regulate oligodendrogenesis and myelination and may act in an intercellular context. We have therefore retained and expanded on the role of TGF- β signaling, highlighting how microglial depletion alters oligodendrocyte immune profiles and how these pathways contribute to myelin maintenance via glia crosstalk. The paragraph has been revised accordingly to more clearly reflect these points.

4. The ordering of physiological and pathological states is somewhat inconsistent in different sections. For example, in the discussion on immune signals regulating microglia-dependent phagocytosis, AD-related changes are described first, followed by physiological

mechanisms. However, in other parts of the review, the physiological role is introduced first, followed by disease-related alterations. Maintaining a consistent structure-either "health to disease" or "disease to health"-throughout the manuscript would improve readability and logical flow.

Our intention throughout the manuscript was to first describe physiological signalling mechanisms, followed by how these are dysregulated in pathological contexts. For example, in the section discussing astrocyte coordination of microglia-mediated synapse pruning, we first describe the role of IL-33, ATP, and complement in development, before moving on to how these pathways are altered in AD. We have reviewed the manuscript carefully and believe this "health-to-disease" structure is consistently applied. However, if there are specific instances the reviewer found confusing, we would be happy to revise those sections for improved clarity.

5. The summary tables provide valuable details on secreted factors across multiple disease models, while the main text primarily focuses on AD. It may be helpful to briefly discuss whether these molecules are specific to AD pathology or part of broader neurodegenerative mechanisms. This additional context would allow readers to better understand the generalizability of these findings and their potential relevance to other neurological disorders.

We agree. Our intention is to highlight the role of this inter-glia network in both health and disease. While we have used Alzheimer's disease (AD) as a central example, many of the pathways and molecules discussed are relevant to a broader range of neuroinflammatory and neurodegenerative disorders. To make this clearer, we have added the following sentence to the introduction of the section "astrocyte-mediated coordination of inter-glia immune signalling":

"While we focus on AD in this review, many of the molecules listed in Table 1 are implicated more broadly in neuroinflammation, a hallmark pathology of various neurodegenerative diseases. It is therefore likely that the pathways discussed here are relevant across a range of neuropathologies beyond AD."

Additionally, the conclusion now includes the following text to emphasise the broader implications:

"Although, it should be noted that this review also highlights convergent pathways in other neuro-pathologies. Wherein, disease-associated profiles of glia are often shared and

this offers insights into the universality of glia as well as potential underexplored pathways for the AD field. This review has also highlighted key limitations in our existing understanding of macroglia communications.”

Referee #2:

We thank the reviewer for highlighting that our review *comprehensively describes the previously underappreciated immunomodulatory role of astrocytes and oligodendrocytes.*

1. Regarding the titles "Astrocytes initiate and engage in immunological signalling pathways" and "Oligodendrocytes initiate and engage in immunological signalling pathways", it might not be very precise to describe astrocytes or oligodendrocytes as the cells that initiate immunological signalling pathway, in a complex multi-cellular system, it's difficult to dissect out which cell types are the initiators, unless in a very specific context.

Thank you to the reviewer for this helpful suggestion. We have changed the subheadings to better reflect a network approach and emphasis intercellular communication more broadly.

‘Astrocyte-mediated coordination of inter-glial immune signalling’

‘Oligodendrocyte-mediated coordination of inter-glial immune signalling’

2. The authors discuss "Astrocytic coordination of microglia-dependent synapse pruning in physiology", some recent studies in the field, e.g. PMID: 39762658, have challenged the previous "established" role of microglia in synapse pruning and circuit formation during brain development, showing normal neuron development without microglia in the brain. These new findings are important, should be discussed or mentioned in this session.

The reviewer makes a good point that recent studies challenge the previously well-established role of microglia in developmental synapse pruning. The finding that neuronal circuits can develop normally in the absence of microglia raises an important question about redundancy and compensation within the glial network. We feel this highlights the capacity of other glial cells, particularly astrocytes, to take on phagocytic roles when microglia are absent. Prior work has shown that astrocytes can phagocytose synapses (PMID:24270812) and it is accepted they have the capacity to act as non-professional phagocytes. To address this, we have added a discussion of this emerging evidence to the end of the section title “Astrocytic coordination of microglia dependent synapse pruning in development” on page 9.

Dear Dr Arancibia-Carcamo,

Re: JP-TR-2025-287015R1 "**The Glia-Immune Network: Astrocytes and Oligodendrocytes as microglial coordinators in health and disease.**" by Verity Fay Toulouse Mitchener, Millie Jennifer Thackray, and I. Lorena Arancibia-Carcamo

We are pleased to tell you that your paper has been accepted for publication in The Journal of Physiology.

Authors should note that it is too late at this point to offer corrections prior to proofing. Major corrections at proof stage, such as changes to figures, will be referred to the Editors for approval before they can be incorporated. Only minor changes, such as to style and consistency, should be made at proof stage. Changes that need to be made after proof stage will usually require a formal correction notice.

Yours sincerely,

Nathan Schoppa
Senior Editor
The Journal of Physiology

P.S. - You can help your research get the attention it deserves! Check out Wiley's free Promotion Guide for best-practice recommendations for promoting your work at www.wileyauthors.com/eeo/guide. You can learn more about Wiley Editing Services which offers professional video, design, and writing services to create shareable video abstracts, infographics, conference posters, lay summaries, and research news stories for your research at www.wileyauthors.com/eeo/promotion.

IMPORTANT NOTICE ABOUT OPEN ACCESS: To assist authors whose funding agencies mandate public access to published research findings sooner than 12 months after publication, The Journal of Physiology allows authors to pay an Open Access (OA) fee to have their papers made freely available immediately on publication.

You can check if your funder or institution has a Wiley Open Access Account here: <https://authorservices.wiley.com/author-resources/Journal-Authors/licensing-and-open-access/open-access/author-compliance-tool.html>.

EDITOR COMMENTS

Reviewing Editor:

Many thanks for revising the manuscript and addressing comments raised by the reviewers.

Senior Editor:

Congratulations! You addressed the prior concerns that were raised in your revision and your topical review is now acceptable for publication.